# LEAP Motion Technology and Psychology: A Mini-Review on Hand Movements Sensing for Neurodevelopmental and Neurocognitive Disorders

**DOI:** 10.3390/ijerph18084006

**Published:** 2021-04-11

**Authors:** Giulia Colombini, Mirko Duradoni, Federico Carpi, Laura Vagnoli, Andrea Guazzini

**Affiliations:** 1Department of Education, Literatures, Intercultural Studies, Languages and Psychology, University of Florence, 50135 Florence, Italy; giulia.colombini@stud.unifi.it; 2Department of Industrial Engineering, University of Florence, 50121 Florence, Italy; mirko.duradoni@unifi.it (M.D.); federico.carpi@unifi.it (F.C.); 3Pediatric Psychology, Meyer Children’s Hospital, Viale Pieraccini 24, 50139 Florence, Italy; laura.vagnoli@meyer.it; 4Centre for the Study of Complex Dynamics, University of Florence, 50121 Florence, Italy

**Keywords:** LEAP Motion, hand movement, virtual reality, neurodevelopmental disorders, neurocognitive disorders, attention-deficit hyperactivity disorder, dementia, mild cognitive impairment

## Abstract

Technological advancement is constantly evolving, and it is also developing in the mental health field. Various applications, often based on virtual reality, have been implemented to carry out psychological assessments and interventions, using innovative human–machine interaction systems. In this context, the LEAP Motion sensing technology has raised interest, since it allows for more natural interactions with digital contents, via an optical tracking of hand and finger movements. Recent research has considered LEAP Motion features in virtual-reality-based systems, to meet specific needs of different clinical populations, varying in age and type of disorder. The present paper carried out a systematic mini-review of the available literature using Preferred Reporting Items for Systematic Reviews and Meta-analysis (PRISMA) guidelines. The inclusion criteria were (i) publication date between 2013 and 2020, (ii) being an empirical study or project report, (iii) written in English or Italian languages, (iv) published in a scholarly peer-reviewed journal and/or conference proceedings, and (v) assessing LEAP Motion intervention for four specific psychological domains (i.e., autism spectrum disorder, attention-deficit/hyperactivity disorder, dementia, and mild cognitive impairment), objectively. Nineteen eligible empirical studies were included. Overall, results show that protocols for attention-deficit hyperactivity disorder and autism spectrum disorder can promote psychomotor and psychosocial rehabilitation in contexts that stimulate learning. Moreover, virtual reality and LEAP Motion seem promising for the assessment and screening of functional abilities in dementia and mild cognitive impairment. As evidence is, however, still limited, deeper investigations are needed to assess the full potential of the LEAP Motion technology, possibly extending its applications. This is relevant, considering the role that virtual reality could have in overcoming barriers to access assessment, therapies, and smart monitoring.

## 1. Introduction

Growing attention has been given to technology-based tools, and researchers are increasingly analyzing their potential to contribute to mental health services [1]. Recently, different technologies have been included in mental healthcare delivery, and this has promoted a reflection on innovative care models that can reach people who might not have access to services [2]. Studies in this field also shed light on the recently developed LEAP Motion technology. The LEAP Motion controller is a highly compact and affordable USB motion capture device with two cameras and three infrared LEDs (Figure 1—left side). Thanks to the illumination of the surrounding space, the device captures hand gestures at a one-meter distance with a mean accuracy of 0.7 mm [3]. A tracking algorithm allows us to estimate the position and orientation of hands and fingers that are directly visible in a three-dimensional virtual representation [4]. In this way, data coming from the LEAP Motion controller allow users to interact within a virtual environment in a touchless way, by using natural hand gestures as input commands [4] (Figure 1—right side).

The LEAP Motion Software Development Kit recognizes simple movements such as swipe, tapping, grabbing, and circular gestures, making it possible to manipulate virtual objects by grasping and placing them [4]. It must be noted that the tracking quality can be altered by too strong or poor illumination of the room and that occluded parts of the hand cannot be traced by the device, even if it can estimate conventional movements [4].

Touchless interaction with small hand gestures could offer opportunities for people with disabilities [5]. Indeed, this kind of user interface is broadly used in gaming but also in assistive technologies, as they are able to identify movements of the body, thus valuable for people with impairments that prevent them from using touch interfaces [6].

Moreover, research also shows the benefits of gesture interaction in populations with developmental disorders, thanks to the possibility to promote motor skills as well as cognitive and social ones in monitored virtual environments that can reproduce real settings [7].

For these reasons, motion capture systems, such as Microsoft’s Kinect, have already demonstrated their usefulness in supporting physical rehabilitation [8,9] and intervention in clinical populations with specific needs [10,11]. However, such systems typically do not allow for the development of low-cost custom applications. The LEAP technology can overcome this limitation, by enabling immediate communication with freeware graphics engines. This has led researchers worldwide to develop a whole series of activities ex novo [12,13,14,15,16].

This technology is easily accessible by populations with different levels of technological expertise and could be used for gamified activities, which are appreciated, for instance, by children [17].

In general, playing, recreational programs [18,19], and virtual reality (VR) activities are often used by hospitals to support people in reducing their fear, distress, and the intensity of perceived pain in various medical procedures [20,21,22]. However, virtual gamified activities can not only be useful for distraction but also can offer a means to assess some psychological dimensions of users [23]. For instance, the use of virtual reality has recently been proposed to battle social isolation in institutionalized elderly people in residential structures, with positive effects regarding the reduction in loneliness [24].

With specific regard to the LEAP Motion technology, researchers have used it to project and implement interventions for neurodevelopmental and neurocognitive disorders that are of interest in this paper.

Neurodevelopmental disorders are a group of disorders characterized by the disorder onset in the developmental period. Indeed, the disorders often manifest before entering grade school, and they are defined by developmental deficits that cause impairments of personal, occupational, social, and academic functioning. Instead, the neurocognitive disorders include disorders characterized by core clinical deficits in cognitive functions. They are not developmental deficits but acquired, indeed, the cognition impairment is not present from birth or very early life, it rather constitutes a decline from a previous level of functioning [25].

Among neurodevelopmental disorders there are autism spectrum disorder and attention-deficit/hyperactivity disorder. Autism spectrum disorder (ASD) is characterized by persistent deficits in social communication and interaction skills as well as repetitive and restricted behavior patterns, interests, and activities. Communication and interaction impairment are shown in different contexts including socio-emotional reciprocity, nonverbal communicative behaviors, and in developing, understanding, and maintaining relationships. Stereotypy can be found in motor movements, use of objects, and speech; inflexible adherence to routines and hyper- or hypo-reactivity to sensory input are other characteristics. The spectrum integrates four pervasive developmental disorders that were considered distinct diagnoses in the DSM-IV: Asperger’s disorder, autistic disorder, childhood disintegrative disorder, and pervasive developmental disorder are not otherwise specified. Prevalence in the U.S. and non-U.S. countries is around 1% of the population [25]. Attention-deficit/hyperactivity disorder (ADHD) is characterized by persistent symptoms of inattention, impulsivity, and/or hyperactivity that interfere with functioning. Inattention may manifest in having difficulty sustaining focus, straying from activities, and being disorganized, for example. Impulsivity is defined by precipitous actions realized without forethought and potentially hurting the person. It may display in deciding without considering consequences and having socially intrusive behaviors. Hyperactivity is shown with excessive and inappropriate motor activity, resulting in extreme restlessness or also talkativeness. In the general population, attention-deficit/hyperactivity disorder is more frequent in boys than in girls. ADHD seems to occur across cultures in about 5% of children and about 2.5% of adults [25].

Among neurocognitive disorders there are major and mild neurocognitive disorders. The major neurocognitive disorder is introduced in DSM-5 as an alternative term to dementia. It is characterized by a significant cognitive decline in one or more cognitive domains including complex attention, learning, language, memory, executive function, perceptual–motor, or social cognition. The cognitive deficits interfere with independence in everyday activities for which the person needs assistance, at least in complex instrumental ones. The maintenance of independent functioning distinguishes the mild and major neurocognitive disorders. Indeed, the mild neurocognitive disorder is characterized by a modest cognitive decline in the same cognitive domains, but cognitive impairment does not interfere with independent functioning in everyday activities [25]. Here, daily tasks become more laborious, and the person needs compensatory strategies [26]. Mild neurocognitive disorder represents a framework for the commonly used diagnosis of mild cognitive impairment (MCI) [26]. Estimates of prevalence for dementia—congruent with major neurocognitive disorder—are about 1–2% at 65 years and 30% by 85 years, while for mild cognitive impairment—congruent with mild neurocognitive disorder—are variable, from 2 to 10% at 65 years and 5 to 25% by 85 years [25].

Authors are also working to define guidelines to develop applications for literacy difficulties in developmental coordination disorder [27,28,29]. Besides, it has also been used for assessment and rehabilitation in Parkinson’s disease [30], cerebral palsy [31], and stroke [32], but here, studies focused particularly on motor areas and physical therapies, so they are not of interest in this paper.

The aim of this mini-review is to provide an overview of existing applications of LEAP Motion for different psychological domains. Specifically, we will describe their implementation and basis for interventions in neurodevelopmental and neurocognitive disorders. Indeed, this review includes studies on attention-deficit hyperactivity disorder and autism spectrum disorder, which are considered neurodevelopmental disorders [33], and dementia and MCI, which are neurocognitive disorders [26].

## 2. Methods

### Search and Selection Strategy

Our mini-review was carried out by using Preferred Reporting Items for Systematic Reviews and Meta-analysis (PRISMA) guidelines. First, we proceeded in searching for scientific studies about LEAP Motion applications in the following four psychological domains: attention-deficit hyperactivity disorder, autism spectrum disorder, dementia, and mild cognitive impairment. The authors accomplished their task using the EBSCO host platform and consulting the databases of PsycInfo, PubMed, Science Direct, Sociological Abstracts, PsycArticles, and Academic Search Complete. The authors also searched in Google Scholar to increase the chances of identifying the widest range of possible sources. Search terms were “Leap Motion” and “ASD”, “Leap Motion” and “ADHD”, “Leap Motion” and “dementia”, “Leap Motion” and “MCI”.

The inclusion criteria were (i) publication date between 2013 and 2020, (ii) being an empirical study or project report, (iii) written in English or Italian languages (the two languages spoken by the authors), (iv) published in a scholarly peer-reviewed journal and/or conference proceedings, and (v) assessed LEAP Motion intervention for the four psychological domains. The search started on January 2020 and ended in August 2020.

Finally, all the sources were merged in a single database, and the duplicates were removed. Of the 865 results obtained during the screening phase, only 71 mentioned “LEAP Motion” (with or without capitalization) together with attention-deficit hyperactivity disorder, autism spectrum disorder, mild cognitive impairment (or their respective acronyms, ADHD, ASD, MCI), or dementia in the title, abstract, or keywords and, thus, were eligible for full-text assessment. Among the 71 results, 52 were excluded, based on the following exclusion criteria for a work: (a) it did not directly test or review LEAP Motion upon or for the target populations; (b) it encompassed physical rehabilitation only; (c) it did not assess the intervention effectiveness on psychological dimensions; (d) it was written in languages other than English or Italian. Finally, it was possible to identify 19 peer-reviewed publications that described LEAP Motion applications in the four psychological domains.

The flow diagram of the study is shown in Figure 2.

## 3. Results

### 3.1. Characteristics of the Studies

Table A1 shows the characteristics of the selected reports. All nineteen studies used a gamified approach. Eleven studies proposed one task, three studies proposed two tasks, three studies proposed three tasks, one study proposed four tasks, and one study proposed seven tasks. Overall, the proposed tasks can be categorized as follows: matching games whose aim is to correctly associate items [34,35,36,37,38,39,40,41]; daily routine games whose aim is to exercise in tasks such as activities of daily living, shopping, greeting, drawing, evacuating by fire, signs recognizing, eye gazing [23,39,40,41,42,43,44,45,46,47]; collaborative games whose aim is to cooperate to complete some tasks [15,16]; mathematical games whose aim is to correctly perform arithmetical operations [48]; labyrinth games whose aim is to correctly reach the end of the path [14].

In general, used stimuli include pictures, words, numbers, and avatars. Among studies involving one task, one study used a matching game with geometric pictures stimuli [34]; one study used a mathematical game with numerical stimuli [48]; one study used a daily routine game with avatar stimuli [42]; two studies used a matching game with picture stimuli [35,36]; four studies used a daily routine game with picture stimuli [43,44,46,47]; one study used a daily routine game with picture and word stimuli [45]; one study used a labyrinth game with picture stimuli [14].

Among studies involving two tasks: two studies used two matching games with picture stimuli [37,38]; one study used a daily routine game and a matching game with picture and word stimuli [39]. Among studies involving three tasks: one study used two daily routine games with picture stimuli and a matching game with picture and word stimuli [40]; two studies used collaborative games with picture stimuli [15,16]. The study with four tasks used two matching games with picture stimuli, and two daily routine games with picture stimuli [41]. The study with seven tasks used daily routine games: three tasks with picture stimuli, two tasks with numerical stimuli, one task with picture and word stimuli, and one task with picture, numerical, and word stimuli [23].

Selected reports included a total of 57 children with ASD (out of which one in mild range, two in moderate range, five in severe range, two in severe range and with mild intellectual disability, one also with ADHD, five in highly functioning range, four in low functioning range); two children with similar characteristics as children with autism (out of which one with better motor skills but focus issues, and one with motor impairments); one child with Down’s syndrome; one child with moderate intellectual disability; 10 children with ADHD; 23 cognitively impaired participants; 65 elderly with amnestic single-domain MCI; 42 elderly with amnestic multi-domain MCI; another 65 elderly with amnestic MCI without specification of single- or multi-domain; 113 elderly with mild Alzheimer’s dementia; 180 healthy elderly; 10 healthy adults; 18 typically developing children (TD); 10 healthy children; 16 elderly with unspecified diagnosis; 19 children with unspecified diagnosis. One study did not specify the number of typically developing children involved [42]. One study did not have participants [45].

Among nineteen studies, eleven studies included both women and men, two studies included only men, no studies included only women, and five studies did not report information about gender distribution. One study did not have participants, and so, gender was not reported.

All studies included only children or only adults or only the elderly. Overall, children’s samples had an age range of 6–12 years, while the elderly one was of 65–85 years. Adults’ age range is not reported.

Among nineteen reports, six studies used one control group. Out of these, four studies used one experimental group, one study used two experimental groups, and one study used three experimental groups.

Among four studies with one experimental group, one study compared children with ADHD to healthy children; one study compared participants with cognitive impairment to participants without cognitive impairment; one study compared ASD/TD children couples with TD/TD children couples; one study compared ASD/TD children couples both in control and experimental groups.

The study that used two experimental groups compared elderly with amnestic MCI and elderly with mild Alzheimer’s disease with healthy elderly.

The study that used three experimental groups compared elderly with amnestic single-domain MCI, elderly with amnestic multi-domain MCI, and elderly with mild Alzheimer’s dementia with healthy elderly.

The other eight studies used one experimental group without control groups, and another one used two experimental groups without control groups.

Furthermore, three studies did not have groups, since two of them used a single subject research design, and one of them used a multiple probe design across participants.

One study did not have participants.

Where specified, the overall duration of performing the task is on average equal or less than 20 min, and a variable number of sessions is carried, ranging from two to twenty.

### 3.2. Study Results

Table A2 shows the main findings of selected reports. Regarding studies with matching games, six studies used picture stimuli for children with ASD. Two studies reported a considerable improvement in fine motor skills and recognition in children with ASD [37,38], and one study observed that Leap-Motion-aided VR technology was more effective in teaching visual matching skills to students with ASD compared to teacher-implemented instructions [36].

Four studies reported improvements in response accuracy in children with ASD [35,37,38,41]: two studies reported 100% accuracy in performing the task after an intervention of half an hour a day [37,38] for five days a week for three weeks [37]; one study reported 11.16 and 16.6% accuracy increase over three training sessions [41]; one study found a functional relationship between gesture-based instruction via Leap-Motion-aided VR technology and response accuracy after 20 pre-experimental training trials every day, with sessions of 10–15 min, and 5 s to provide response in the intervention phase [35], while another study reported variable accuracy percentages under Leap-Motion-based computer-assisted instructions (CAI) and teacher-implemented instructions (TII) interventions after 20 pre-experimental training trials every day for each intervention, with sessions of 10–15 min, and 5 s to provide response in the intervention phase [36]. One study observed 10.67% improvement in children with ADHD after an average playtime of 16.56 min across three attempts made in a week [34]. Capelo et al. [34] also reported an increase in children’s relaxation, motivation, and concentration. Two studies reported the promotion of task engagement with the Leap-Motion-based CAI approach [35,36].

One study found maintenance of acquired skills at a high level up to 12 weeks under CAI [35], while another study observed maintenance at a high level up to 5 weeks under both CAI and TII [36].

Two studies reported generalization and transfer of learned skills [36,37]. Two other studies used matching games with word and picture stimuli [39,40]. Both reported high levels of sustained attention and engagement in children with ASD as well as an increase in independent manipulation.

Regarding daily routine games, five studies used picture stimuli for elderly with cognitive impairment and two studies for children with ASD. Among those that addressed the elderly, two studies reported that total virtual measures of functional abilities showed consistent functional impairment in the experimental groups if compared with control group [46,47]. Indeed, both studies showed that the LEAP-Motion-aided VR technology performance can discriminate between cognitive impaired participants and cognitive intact elderly. One of them found a strong correlation between the virtual-assessed functional index and two standard cognitive and functional measurement scales scores (i.e., Mini-Mental State Examination and Bristol Activities of Daily Living scale) [47]. Besides, one study reported an analogy between patient clustering obtained by using acceleration data coming from LEAP-Motion-based activity, and clusters formed thanks to performance measures [43].

A study used daily routine games with children with ASD and proposed avatar stimuli. The authors observed that ASD children were able to learn promptly from the proposed activity (i.e., in just 20–30 s of the first training session), thus improving their communication skills [42].

Three other studies used daily routine games and proposed word and picture stimuli. Two studies involved children with ASD and found high degrees of sustained attention, task engagement, and enjoyment after a playtime of 15 min [39,40]. From one study addressing elderly with cognitive impairment, the author’s reporting found limitations in Leap Motion usage in daily routine games due to accuracy issues and light influence [45].

Finally, one study used daily routine games proposing all different kinds of stimuli (i.e., picture stimuli, numerical stimuli, picture and word stimuli, and picture, numerical, and word stimuli) for elderly with cognitive impairment. It reported a moderate positive correlation between the total performance scores and three validated cognitive screening tools scores (i.e., Abbreviated Mental Test, Mini-Mental State Examination, and Montreal Cognitive Assessment) and a moderately significant relationship between the total performance scores and the presence of cognitive impairment [23]. In this study, outcomes were obtained after an average time of 20.4 min (s.d. = 3.4) to complete the task in the experimental group and an average time of 19.1 min (s.d. = 3.6) in the control group.

Furthermore, two studies used collaborative games with children with ASD. One study reported that, on average, playtime tended to decrease over the experimentation period, while participants’ collaborative efficiency increased. This result was found for both experimental and control groups. [15]. Specifically, an improvement of 5.49% in collaborative efficiency was reported for the experimental group and of 20.64% for the control one. These outcomes were reported after a total playtime of 5 min approximately. Another study reported improvements in cooperation and communication in the experimental group as well as an increase in the number of words spoken per minute by children with ASD [16]. Both studies observed an increase in spontaneous communication.

Moreover, one study used a mathematical game for children with ADHD. It reported meaningful correlations between the scores attributed to the interface and the children’s learning outcomes; it also found an improvement in attention [48].

Finally, one study used a labyrinth game for children with ASD. It reported a high percentage of agreement among expert therapists about the training of children’s focus [14].

### 3.3. Risk of Bias

Table A2 shows the main risks of bias within the selected reports. Nine studies did not report sampling criteria [14,15,16,34,39,40,41,44,48]. The possible absence of eligibility criteria may have induced a biased recruitment process. Furthermore, five studies did not report any information about blinding [14,34,41,43,48]. The possible absence of blinding could have had an influence on the outcomes. In two studies, participants were informed on the nature of the game beforehand [39,40]. This could have modified participants’ performance. Two studies used a non-probability sampling technique based on recommendation [39,40]. A nonprobability sampling of this kind may have induced a biased recruitment process. Moreover, two studies did not include a clinical sample [42,43]. One study included a sample with a diagnosis diverse from the targeted one [41]. Two studies did not specify sample diagnosis [44,48], and thus, their results are not easily generalizable to the target population. Besides, two studies used a male-only sample [35,39]. Two studies excluded participants with technophobia from the sample [46,47]. One study was unable to include very young children in the sample because of LEAP Motion accuracy issues [36]. One other study reported technical issues with LEAP Motion with older children [15]. Therefore, gesture-recognition problems could have affected the results. One study sampled the study population relying on scores from a non-diagnostic tool [23]. Participants with a normal score but a subjective cognitive impairment may have been wrongly included in the control group. One study used an assessment module that was originally used for rehabilitation and then adapted for cognitive screening [23]. Tasks included in the assessment module may have been difficult to deal with not because of the participants’ cognitive impairment, but due to the prototype content design. Moreover, in two studies, the authors claimed that they used statistical models with a limited number of covariates, and they may have omitted some important confounders [46,47]. This could have modified the outcomes. Finally, two studies used a single subject research design [37,38].

## 4. Discussion

### 4.1. Distribution of Studies on LEAP Motion Applications in the Various Domains

The analyzed 19 studies [14,15,16,23,34,35,36,37,38,39,40,41,42,43,44,45,46,47,48] have shown that the LEAP Motion sensing technology is typically combined with virtual environments, in order to implement interactive and immersive video games with specific tasks targeting possible interventions on different clinical populations. The tasks aim to evaluate and/or enhance deficient skills in interventions for various psychological domains.

Specifically, two studies consider children with attention-deficit hyperactivity disorder [34,48], 11 studies address children with autism spectrum disorder [14,15,16,35,36,37,38,39,40,41,42], while six consider adults with dementia or mild cognitive impairment [23,43,44,45,46,47].

### 4.2. Objectives of LEAP-Motion-Based Interventions

Psychological interventions performed so far with LEAP Motion have had different specific purposes, depending on the nature of the clinical condition considered. However, recurring objectives have typically included the evaluation and/or the enhancement of deficient areas.

Protocols for neurodevelopmental disorders have been aimed to promote, above all, psychomotor and psychosocial rehabilitation in contexts that stimulate learning. Concerning ADHD, interventions have targeted training of sustained and focalized attention, as well as hand–eye coordination [34,48]. This is because ADHD is characterized by an attention-deficit, often linked to fine motor impairments and visuo-spatial skills difficulties, which also have consequences on the learning process [34].

Similarly, ASD interventions have been focused on the improvement of fine motor skills and visual motor integration, fostering attention and motor control [14,37,38,39,40,41], as these functions have been found to be commonly problematic [37]. Learning difficulties have been supported too [35,36]. Additionally, socialization, communication, and independence have been encouraged by specific interventions [15,16,42], given the persistent deficits displayed.

Interventions for neurocognitive disorders typically have targeted cognitive screening, assessment of the impairment, and cognitive rehabilitation. In dementia, the core goals have concerned the evaluation of executive functions and the exercise of everyday activities associated with memory stimulation [44,45,47]. Likewise, in MCI, the cognitive performance has been assessed, and the action impairment has again been the focal point of intervention [23,43,46]. Indeed, declines in these domains are considered defining characteristics of this kind of disorder [49].

### 4.3. Protocols of LEAP-Motion-Based Interventions

Interactivity, immersivity, and multi-sensory stimulation are keywords in designing interventions for both neurodevelopmental and neurocognitive disorders. Indeed, LEAP Motion has been introduced in gamified virtual environments for engaging users to complete particular tasks, specifically implemented to assess or strengthen impaired functions. The following sections describe the protocols.

Protocols in neurodevelopmental disorders: ADHD and ASD.

Protocols in neurodevelopmental disorders have been based on the gamified manipulation of virtual objects and multisensory learning.

Hand–eye coordination and fine motor skills have been among the pivotal areas targeted in studies on ADHD. Garcia-Zapirain et al. [48] addressed them in a dual system for the rehabilitation of cognitive functions of children. Using an eye-tracker and LEAP Motion, participants could interact with an arithmetic gamified application and perform operations with numbers displayed on virtual flower’s petals. Users could introduce the correct solution using the eye gaze and the hands, by stretching the same number of fingers as the number of the result. The main outcomes showed that this hand–eye coordination exercise helped to improve users’ skills and attention, whereas the natural interaction devices proved to be engaging alternatives to handwriting or other kinds of interfaces. The underlying idea is that learning requires the interaction of different sensory modalities with activities that stimulate not only visual analysis and cognition, but also physical movements. This is also suggested by Capelo et al. [34] who used LEAP Motion in a multisensory virtual game, in which participants had to place different geometric figures (i.e., blocks, cubes, spheres) in color-matching containers. The authors found an increase in concentration and motivation levels, in a natural and entertaining interaction. Besides, the game promoted relaxation when children interacted with LEAP Motion because the device turned out to be easy to use.

Zhu et al. [38] implemented two similar LEAP-Motion-based games for children with ASD. In the first one, the task was to grasp and put some balls in boxes of the same color, while in the second one, users had to match fruits to some sticks. Despite the small sample size, the authors reported an improvement in fine motor skills and recognition, with the achievement of 100% accuracy in completing the task. Cai et al. [37] replicated the procedure and confirmed the aforementioned enhancements, underlining also a learning transfer of skills and rules. In particular, abilities such as looking at the hands and objects and moving the gaze with them were increased by the game. The author attributed this result to a probable combination of comprehension of the task rules together with the improvement of fine motor skills and recognition in interacting with LEAP Motion. As stated by the authors, this is one of the early attempts to investigate the effect of using gesture-based games for developing such skills in children with ASD.

Furthermore, even Tang et al. [39] considered LEAP Motion as a useful tool in order to train fine motor skills, especially because of its portability. In a first pilot study, they proposed a drawing game [39], whereas in a second study, they investigated an interaction with a domestic environment and in a zoo thanks to a word–image pairing task [39]. The results underlined high levels of sustained attention and showed that engagement with stimulating tasks, which was noticed with minimum training, made children practice specific movements, allowing them to develop motor control and learning towards more complex motor patterns. The authors also reported that parents and caregivers were involved, they noticed their child’s enduring attentiveness and commitment, and this could probably increase the possibility to extend the training at home, contributing to consolidating the learning.

Recently, Tang et al. [40] introduced LEAP Motion in a drum-playing game, observing that it can promote an entertaining learning approach. They also found that the acceptability of the application depended on the task being natural and that the children’s engagement was not influenced by the severity of the disorder.

Syahputra et al. [14] tried to train attention and focus relying on the abilities of children to move virtual objects, in particular collecting coins, while manipulating an item in four labyrinths of increasing difficulty. As a result, LEAP Motion was deemed able to exercise focusing in children.

Likewise, Rahmadiva et al. [41] addressed the focus of children with ASD and their social skills. They described multiple games, including a color-matching game with balls and boxes; a similar one, with fish and containers in a virtual underwater world; an activity of movement across virtual streets following signs and signals to meet virtual people; a game of item selection according to the gaze direction of a virtual character. The results indicated that participants were engaged and that LEAP Motion could be employed as a device in virtual settings for children with autism, even if its use as a means of rehabilitation requires practice.

To teach visual matching skills to students with ASD, Hu et al. [36] proposed an innovative LEAP-Motion-based computer-assisted instruction (CAI) approach. This was compared to a traditional teacher-implemented instruction (TII) approach in a task of daily item matching. Results showed that the innovative CAI was more effective in teaching the target skills to students with ASD, it showed to be more engaging, and some participants achieved a higher level of accuracy during the intervention with it. The authors concluded that it could promote their independence and learning. Hu and Han [35] investigated the same procedure and confirmed the aforementioned outcomes, underlining high task engagement and the maintenance of the acquired skills for three months. Nevertheless, Hu et al. [36] also observed some low accuracy issues because the hand-gesture recognition showed drops with young children’s small hands.

Other studies have been focused on social skills, developing collaborative virtual environments (CVE) with LEAP Motion for children with ASD. Zhao et al. [15,16] designed a series of collaborative games, aiming to foster socialization and communication. Specifically, they implemented a puzzle, a collection, and a delivery game, which required two users to spend an equal and coordinated effort, in order to match, move, and place virtual objects, usually across obstacles. To complete the task, they had to control a virtual tool with two handles, designed for a natural and more immersive experience. Results showed improvements in children’s engagement and motivation, as well as an increase in cooperation patterns and growing spontaneous communication.

Halabi et al. [42] included LEAP Motion and other devices in a virtual-reality-based system aiming to improve the social performance of children with ASD. They proposed a virtual school setting that involved the user in greetings and conversations with a teacher avatar. Usability studies showed that the system had a positive impact on communication skills.

### 4.4. Protocols in Neurocognitive Disorders: Dementia and MCI

Protocols in neurocognitive disorders have been based on the gamified simulation of basic behaviors and functional abilities, including personal living.

The pioneering use of LEAP Motion for dementia is described in a pilot study by Tarnanas, Schlee et al. [47]. They employed it together with other devices to collect information about the rate of change in users’ functional impairment in a task of fire evacuation of a virtual apartment, designed with different scenarios of growing difficulty. The authors aimed at improving the ecological validity of such measures as a screening tool for early dementia. The participants had to move on a treadmill to approximate the actual movements in front of a projection screen, and they could interact with the environment thanks to hand gestures (i.e., Leap Motion and Kinect), planning a strategy to evacuate safety. The results showed that virtual reality, motion tracking, and natural tasks could help in pre-dementia diagnosis, executive function assessment, and intervention.

Tarnanas, Mouzakidis et al. [46] examined the same system; in an activity of the daily living module, the psychomotor evaluation was conducted through performance measures in different tasks, to evaluate the users’ understanding and their abilities to perform specific physical tasks accurately. Here, LEAP Motion was used in a finger-tapping test. The outcomes of the study confirmed the possibility to also contribute to MCI diagnosis, by measuring functional abilities in virtual reality with such devices.

Similarly, Martono et al. [43] considered everyday action impairment in a pilot study. They designed a lunch box packing task with specific steps (e.g., taking bread and spreading jelly, wrapping a sandwich, taking cookies and juice) as daily activity in a virtual kitchen on a tablet. LEAP Motion recorded finger movements, and data were used to create clusters of participants; at the same time, the authors realized a performance-based assessment of each user, describing errors made during the exercise. Despite sample limitations, results from a comparison suggested that this approach could be relevant to cluster patients according to virtually assessed symptoms.

Unlike the aforementioned studies, Sacco et al. [45] identified some limitations of LEAP Motion. In a project of virtual training for visuo-spatial abilities in elderly with minor cognitive disorders, LEAP Motion was compared with other devices in a virtual shopping task. The user had to read the name of some products from a shopping list, to identify the correct lane of the market to find them, and finally, to select the right items on the shelves. LEAP Motion showed to be influenced by light and to have some lack of accuracy, due to the fact that the hand obstructs the tracking when it is perpendicular to the device.

Nevertheless, Chua et al. [23] used virtual reality and LEAP Motion to mimic everyday activities in three-dimensional games, from which to assess not only executive functions, but also memory, perceptual motor skills, and learning. Seven activities were proposed: users had to open a door by means of the right key and a code, to make a phone call typing a number, to identify famous people, groceries advertisement and numbers on a newspaper, to organize house objects in categories, to select an outfit according to a specific occurrence, to take cash from a teller machine, and to do shopping. Preliminary outcomes concluded that virtual reality and LEAP Motion could be used for the screening of cognitive functions in older people in primary care settings.

Vallejo et al. [44] implemented a table preparation task in a virtual kitchen setting, in which the aim was to place some kitchenware accurately and quickly, stimulating executive functions and response time. The author compared the usability of LEAP Motion to another interface (i.e., Razer Hydra) and found that participants preferred the first one, even if they were faster to finish the task using the last one. However, conclusions showed that LEAP Motion represented well the reality of movements such as taking and displacing, and it seemed promising for virtual rehabilitation.

### 4.5. Background for LEAP-Motion-Based Interventions

The reviewed studies have analyzed the use of LEAP-Motion-based gesture interaction systems in interventions for different clinical populations. Protocols were designed to target various psycho-motor functions and psycho-social skills, on the basis of evidence coming from studies in the field of virtual reality and motion-based gaming.

Those studies have shown that games can provide safe environments for practicing in various tasks, as many times as the user needs, and can also improve learning from mistakes owing to a motivating real-time feedback [37] and self-paced activities. Indeed, research indicates that children with ASD can learn how to behave in social settings when they constantly train in specific situations [50], whereas extended practice of everyday activities can enhance performance in people with neurocognitive disorders [43].

Moreover, evidence shows that games can promote improvements in hand–eye coordination and visuospatial skills [51], also encouraging decision making and cognitive strategies [34]. Motion-based gaming can foster learning, attention [52], and psycho-motor skills, due to the continuous need of motor actions [34]. This is relevant in designing game-based protocols given that these areas are particularly important for all the aforementioned clinical populations, especially in ADHD and in ASD.

Other evidence deals with engagement; for instance, the interaction of children with ASD with natural electronic devices show a high involvement with virtual reality [53,54,55,56], making them actively engaged [57,58] and avoiding being overburdened by stimuli as happens in human interactions [54]. This turns out to be useful for rehabilitation interventions and screening.

Furthermore, virtual reality can generate ecological validity [47], involving participants in the task with a minor focus on the testing procedure, contrary to traditional measures [59,60]. As shown in studies on neurocognitive disorders, gesture-based games can also allow for assessing executive functions and perceptual motor functions that often are only partially considered in paper-and-pencil screening tools [61].

In summary, whilst the reviewed studies were based on such previous evidence on the usefulness of virtual-reality-based procedures, they not only confirmed those preliminary findings, but especially showed the potential (as well as the current limitations) of the use of the LEAP Motion technology to that aim.

### 4.6. Technology Weaknesses and Future Challenges

Alongside positive impact outcomes, some technology weaknesses have been reported. Accuracy issues have been noted in LEAP Motion tacking ability; particularly, it has been found to be weaker when the hand is positioned perpendicularly to the device, and to be influenced by light [45]. Moreover, some studies reported negative feedback addressing LEAP Motion lacking sensitivity with younger children’s hands that were too small to be correctly detected [15,36]. This led to their exclusion from the study or to a sub-optimal fit to its usage.

Thence, future modifications would be needed to adjust its abilities in order to also fit with children’s characteristics, thus supporting not only a better game experience but also an improved validity in studies including it.

## 5. Conclusions

This mini-review provided a glimpse on ongoing applications of the LEAP Motion hand tracking technology for interventions in neurodevelopmental disorders, in particular in ADHD and ASD, and in neurocognitive disorders, specifically in dementia and MCI. Across these clinical populations, LEAP Motion has been introduced to interact with gamified virtual environments, designed to engage the user with specific tasks, from which impaired functions can be assessed and/or enhanced.

The use of the device has been shown to have a significant possible impact, especially on interventions targeting improvements of psycho-motor functions and psycho-social skills. The main affected areas in neurodevelopmental disorders are hand–eye coordination, visual matching skills, fine motor skills, sustained and focalized attention as well as concentration and motivation, communication skills, cooperation, and socialization. These domains can be enhanced through an entertaining and multi-sensory learning approach that uses the LEAP Motion technology in matching games, virtual object manipulation, and collaborative activities.

In contrast, LEAP-Motion-based interventions in neurocognitive disorders have been focused on basic behaviors and functional abilities (e.g., everyday activities, executive functions), proposing gamified tasks that stimulate them. The results showed that virtual reality and motion tracking could contribute to pre-dementia and MCI diagnosis and to the clusterization of patients according to virtually assessed symptoms. Therefore, LEAP Motion is considered promising for the screening of cognitive functions in older people but also for early dementia virtual rehabilitation.

In summary, LEAP Motion seems to support different clinical aims. Nevertheless, it is worth stressing that evidence is still limited. This suggests that future studies should be more comprehensive, even through longitudinal methods with many representative samples. To date, research has been performed on small numbers of participants, sometimes of non-clinical kind.

Further research should also provide normative data on different activities, in order to facilitate and regulate the introduction of such approaches into clinical practice. Moreover, future studies should explore the presence of long-term benefits, since just one study reported them. Here, it should also be considered a possible use of LEAP Motion for follow-up studies and re-interventions if necessary, providing evidence about these phases.

Not all studies reported information about the duration and frequency of task performance; therefore, future research could also analyze the impact of these variables on the outcomes.

It is also worth noting that, despite the motion detection ability of LEAP Motion, issues with its accuracy have been reported [15,36,45]. Therefore, technical improvements are desirable, as tracking problems have also been observed with little children that are among the target populations.

Considering the device portability, low cost, and ease of use, additional future applications of this technology are expected in home environments, potentially enhancing home care services [62,63]. Considering the role of virtual reality, this could contribute to overcoming barriers to access assessment, therapies, and smart monitoring. For example, conducting therapeutic sessions at home could help to reduce economic and logistical costs for patients and families. Moreover, studies report that virtual-reality-based interventions provide settings that are similar to games, offering motivating and involving environments, thus allowing a prolonged training session and a better adherence to treatment [64]. Researchers should integrate in future studies training and analysis with caregivers, as they could help in extending interventions with patients, children, or the elderly at home with the device if correctly trained. This is relevant, considering that a wide range of studies supports the mediation of caretakers in interventions that, in this way, can lead to improvements for both patients and caregivers [65]. Here, the compact size and intuitive usage of the controller are valuable and make its application easily accessible for people with different technological expertise.

Future research could also test LEAP Motion to reduce isolation for both children and the elderly at home or in the hospital [66,67]. For instance, studies on stroke show that the sense of isolation can restrict the engagement coming from therapy; for this reason, research in this field has already begun to explore the use of virtual reality and serious gaming with multiple users to support patients [62]. This could also be promoted with LEAP Motion, since it has shown its potential in this area.

## Figures and Tables

**Figure 1 ijerph-18-04006-f001:**
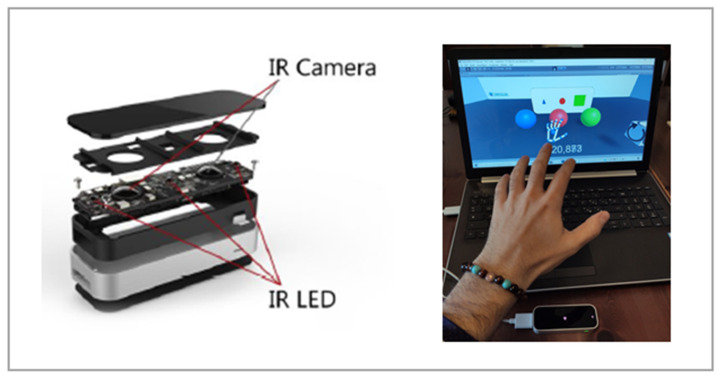
Exploded view of the LEAP Motion device. Reference taken from Wozniak et al., 2016 (left side) and an example of a LEAP-Motion-based virtual environment (right side).

**Figure 2 ijerph-18-04006-f002:**
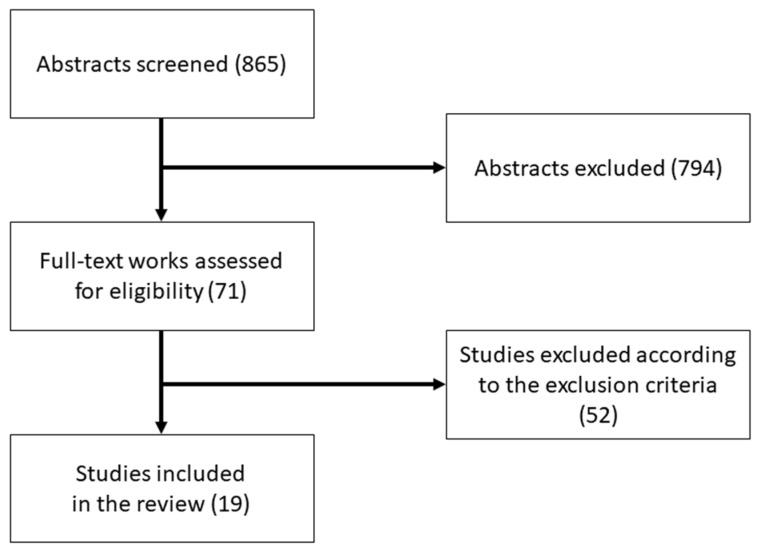
Diagram showing the information flow through the mini-review: the number of records identified, included, and excluded.

## Data Availability

The data presented in this study are available on request from the corresponding author.

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
