# Peer review of "LEAP Motion Technology and Psychology: A Mini-Review on Hand Movements Sensing for Neurodevelopmental and Neurocognitive Disorders"

_ijerph, 2021, doi:10.3390/ijerph18084006_

Round 1

Reviewer 1 Report

This mini-review should be reconsidered, since having carried out a search in the literature and having invested time in it, the scientific character is not reflected. I suggest that the authors rethink the paper following the PRYSMA guide

Abstract:

The summary does not contain any specific information about the review carried out.

Introduction:

It focuses too much on technology and the term psychological domains is hardly described at all.

In the objective it is the first time that the terms are spoken:

 Attention Deficit Hyperactivity Disorder and Autism Spectrum Disorder that are considered neurodevelopmental disorders, and Dementia and MCI that are neurocognitive disorders

They should have been defined and explained more in depth earlier before

  1. Methods

2.1. Search and Selection Strategy

The MESH terms do not appear, nor are the inclusion criteria established, nor when the search was performed.

When this phrase is mentioned, Line 113:

“Finally, it was possible to identify 19 peer-reviewed publications that described LEAP Motion applications in the four psychological domains.”

The bibliographic references of the articles should appear or included

  1. Results

A table of results and more information about the global count of the articles is missing.

Not each article is objectively described, neither the patients, nor the specific interventions.

In this section each article included in this review should be exposed, and in the discussion section the articles should be discussed among themselves.

The section that the authors call results is more of a discussion section

  1. Conclusions

The section of conclusions is inadequate, since according to what was stated in the previous sections, it is difficult to draw conclusions with rigor

Author Response

Dear Reviewer, 

Thank you for sharing the reviews for our submission entitled “LEAP Motion technology and psychology: A mini-review on hand movements sensing for neurodevelopmental and neurocognitive disorders”. We are grateful for having had the opportunity to submit a revised version of the manuscript. Furthermore, we want to express our gratitude for the careful and meticulous reading of the paper. The review was detailed and helpful to improve the manuscript. Below is a point-to-point overview of all changes made to the manuscript and how we have dealt with the suggestions.

Reviewer comment: This mini-review should be reconsidered, since having carried out a search in the literature and having invested time in it, the scientific character is not reflected. I suggest that the authors rethink the paper following the PRYSMA guide

Authors’ answer: Thank you very much for allowing us to improve our paper. As you wisely suggested, we adjusted our paper to match as much as possible PRISMA guidelines. The details will be discussed more specifically in the following points. 

Reviewer comment: Abstract: The summary does not contain any specific information about the review carried out.

Authors’ answer: We totally agreed with you that not specifying this information represents a missed opportunity. Thus, we edited the abstract including a statement about PRISMA guidelines and Inclusion criteria. 

Reviewer comment: Introduction:It focuses too much on technology and the term psychological domains is hardly described at all. In the objective it is the first time that the terms are spoken: Attention Deficit Hyperactivity Disorder and Autism Spectrum Disorder that are considered neurodevelopmental disorders, and Dementia and MCI that are neurocognitive disorders. They should have been defined and explained more in depth earlier before

Authors’ answer: Thank you for highlighting this point. The narration flow appeared impaired by this noticeable absence. For this reason, we have rethought our introduction including an in-depth description of all the disorders addressed by our mini-review. [Lines 85-133]

Reviewer comment: Reviewer comment: Methods 2.1. Search and Selection Strategy. The MESH terms do not appear, nor are the inclusion criteria established, nor when the search was performed.

Authors’ answer: We included in the Method section the information requested [Lines 155-161]

Reviewer comment: When this phrase is mentioned, Line 113: “Finally, it was possible to identify 19 peer-reviewed publications that described LEAP Motion applications in the four psychological domains.”The bibliographic references of the articles should appear or included

Authors’ answer: As you recommended we included at the end of the phrase the bibliographic references of the 19 articles. 

Reviewer comment: Results A table of results and more information about the global count of the articles is missing. Not each article is objectively described, neither the patients, nor the specific interventions. In this section each article included in this review should be exposed, and in the discussion section the articles should be discussed among themselves. The section that the authors call results is more of a discussion section. 

Authors’ answer: Thank you for raising this issue. To meet your requests we revised the paper structure. First, the old results section since it resembled more a discussion has been transformed into a proper discussion. We produced a brand new result section where each article is objectively described together with the patients and specific interventions taking inspiration from this work: https://doi.org/10.3389/fpsyg.2019.00303. To make the results easier to read we also provided in Appendix two dedicated tables (i.e., Table A1, Table A2) about: sample size, gender distribution, age range, mean age, sample characteristics, and task information, main findings, study limitations, and risk of biases. 

Reviewer comment: Conclusions. The section of conclusions is inadequate, since according to what was stated in the previous sections, it is difficult to draw conclusions with rigor. 

Authors’ answer: We hope that by providing the required information by answering the previous points, the conclusion could be now more rigorous. 

Reviewer 2 Report

This mini-review provided an overview of existing applications of the LEAP Motion hand tracking technology for interventions in neurodevelopmental and neurocognitive disorders, in particular in ADHD and ASD, Dementia and MCI. Although the evidence is still limited, LEAP Motion features in virtual reality-based systems has shown its advantages in their implementation and basis for interventions in neurodevelopmental and neurocognitive disorders. This review can inspire and guide researchers to further use the developing information technology for accurate evaluation and intervention of psychological behavior problems.

Author Response

Dear Reviewer, 

Thank you for sharing your comments for our submission entitled “LEAP Motion technology and psychology: A mini-review on hand movements sensing for neurodevelopmental and neurocognitive disorders”. We are grateful for having had the opportunity to submit a revised version of the manuscript. Furthermore, we want to express our gratitude for the careful and meticulous reading of the paper. The review was detailed and helpful to improve the manuscript. Below is a point-to-point overview of all changes made to the manuscript and how we have dealt with the suggestions.

Reviewer comment: This mini-review provided an overview of existing applications of the LEAP Motion hand tracking technology for interventions in neurodevelopmental and neurocognitive disorders, in particular in ADHD and ASD, Dementia and MCI. Although the evidence is still limited, LEAP Motion features in virtual reality-based systems has shown its advantages in their implementation and basis for interventions in neurodevelopmental and neurocognitive disorders. This review can inspire and guide researchers to further use the developing information technology for accurate evaluation and intervention of psychological behavior problems.

Authors’ answer: Thank you very much for your words of appreciation. We do really think that our work could inspire and guide other scientists. Thank you once again for taking the time to carefully read our manuscript.

Reviewer 3 Report

The submitted manuscript entitled: "LEAP Motion technology and psychology: A mini-review on 2 hand movements sensing for neurodevelopmental and neu-3 rocognitive disorders ", numbered by ijerph-1141219 present important issues related to application of existing applications at base of LEAP Motion technology in neurodevelopmental (such as ADHD i ASD) and neurocognitive disorders (such as Dementia and MCI). The paper is interesting and fall within the scope of the Interantional Journal of Environmental Research Public Health

The title and abstract reflect the content of the work.

The work has a review character and base on research other authors, who proved the impact of the virtual environment (games created on the LEAP technology) on psychomotor and cognitive functions in various clinical groups, what is important to cognitive screening, assessment of the impairment and cognitive rehabilitation. The authors reviewed over 800 publications, of which 19 were used to prepare the work.

I think, it will be advantageous to extending the information about the frequency and duration of performing tasks with use of those applications. Although this is not the main issue of the work, I believe that it is important information. How it effects on the observed "positive effects"?

Discussion and conclusions were made correctly based on the current research of other authors and authors rethinks.

The selected topic  is interesting. The work is properly documented and according to my opinion the paper can be published without significant changes.

Strong sides: 865 scientific studies about LEAP Motion applications under these psychological domains were reviewed, the examples of different application were presented (games based on grouping, on matching and skil games).

Weakness sides: not enough details from realised tests other authors, for example: age, sex, extent of disease pepople who work with theses aplication. Number of person of tested, frequency and duration of performing tasks with use of those applications, results (how it extent to helps in this disease).

Author Response

Dear Reviewer, 

Thank you for sharing your comments for our submission entitled “LEAP Motion technology and psychology: A mini-review on hand movements sensing for neurodevelopmental and neurocognitive disorders”. We are grateful for having had the opportunity to submit a revised version of the manuscript. Furthermore, we want to express our gratitude for the careful and meticulous reading of the paper. The review was detailed and helpful to improve the manuscript. Below is a point-to-point overview of all changes made to the manuscript and how we have dealt with the suggestions.

Reviewer comment: I think, it will be advantageous to extending the information about the frequency and duration of performing tasks with use of those applications. Although this is not the main issue of the work, I believe that it is important information. How it effects on the observed "positive effects"? 

Weakness sides: not enough details from realised tests other authors, for example: age, sex, extent of disease pepople who work with theses aplication. Number of person of tested, frequency and duration of performing tasks with use of those applications, results (how it extent to helps in this disease).

Authors’ answer: Thank you very much for this comment that gave us the possibility to further improve the quality of our work.
Unfortunately, information about how much different task durations impacted patients’ improvement is missing (i.e., the studies did not vary this parameter within their experimental sessions). We stressed this point in our conclusion [Lines 597-599]. Nonetheless, we totally agree with you that providing information about the duration of performing tasks is essential. For this reason, we reported the task duration for all the works that provided such data in the results section [Lines 254-266]. Moreover, to remedy the weaknesses indicated by the reviewer we added information about studies’ sample size, gender distribution, age range, mean age, sample characteristics, and task information, main findings, study limitations, and risk of biases in the results section and in two dedicated tables in the appendix (i.e., Table A1, Table A2).

Reviewer 4 Report

Manuscript provides timely interest in mental health diseases. Reviewing specific technologies available at our fingertips would benefit readers with specific curiosity. The method strategy is fair. Given the mini nature of the article, there are certain limitations in reviewing at large. But I would like to see a separate section discussing the current weakness of the technology and future challenges. Followed by the overall conclusion. That would place the manuscript in the perfect organization. 

Author Response

Dear Reviewer, 

Thank you for sharing your comments for our submission entitled “LEAP Motion technology and psychology: A mini-review on hand movements sensing for neurodevelopmental and neurocognitive disorders”. We are grateful for having had the opportunity to submit a revised version of the manuscript. Furthermore, we want to express our gratitude for the careful and meticulous reading of the paper. The review was detailed and helpful to improve the manuscript. Below is a point-to-point overview of all changes made to the manuscript and how we have dealt with the suggestions.

Reviewer comment: Manuscript provides timely interest in mental health diseases. Reviewing specific technologies available at our fingertips would benefit readers with specific curiosity. The method strategy is fair. Given the mini nature of the article, there are certain limitations in reviewing at large. But I would like to see a separate section discussing the current weakness of the technology and future challenges. Followed by the overall conclusion. That would place the manuscript in the perfect organization. 

Authors’ answer: Thank you for your suggestion that actually helped us to organize the paper better. We proceeded, as you recommended, to add a dedicated section about “Technology weaknesses and future challenges” before our overall conclusion [Lines 553-564].

Round 2

Reviewer 1 Report

Congratulations
The article has acquired much more scientific direction and has improved a lot